# HER2 Intratumoral Heterogeneity in Breast Cancer, an Evolving Concept

**DOI:** 10.3390/cancers15102664

**Published:** 2023-05-09

**Authors:** Yanjun Hou, Hiroaki Nitta, Zaibo Li

**Affiliations:** 1Department of Pathology and Laboratory Medicine, Atrium Health Wake Forest Baptist Medical Center, Winston-Salem, NC 28659, USA; 2Roche Tissue Diagnostics, Tucson, AZ 85755, USA; 3Department of Pathology, The Ohio State University Wexner Medical Center, Columbus, OH 43210, USA

**Keywords:** HER2, breast cancer, heterogeneity, immunohistochemistry, digital imaging analysis

## Abstract

**Simple Summary:**

HER2 intratumoral heterogeneity (ITH) is a well-known phenomenon in breast cancer, defined as the coexistence of subpopulations of tumor cells with different HER2 gene or protein expression within a tumor. HER2 ITH has been reported in up to 40% of breast cancers and to be associated with poor prognosis in patients with anti-HER2 targeted therapies and was proposed to be a potential mechanism for anti-HER2 resistance. HER2 ITH can be divided into non-genetic and genetic ITH based on different HER2 genetic amplification and genetic ITH has clustered, mosaic and scattered distribution patterns. HER2 ITH interpretation can be challenging, but digital image analysis has emerged as a potential method to accurately and objectively to assess HER2 ITH.

**Abstract:**

Amplification and/or overexpression of human epidermal growth factor receptor 2 (HER2) in breast cancer is associated with an adverse prognosis. The introduction of anti-HER2 targeted therapy has dramatically improved the clinical outcomes of patients with HER2-positive breast cancer. Unfortunately, a significant number of patients eventually relapse and develop distant metastasis. HER2 intratumoral heterogeneity (ITH) has been reported to be associated with poor prognosis in patients with anti-HER2 targeted therapies and was proposed to be a potential mechanism for anti-HER2 resistance. In this review, we described the current definition, common types of HER2 ITH in breast cancer, the challenge in interpretation of HER2 status in cases showing ITH and the clinical applications of anti-HER2 agents in breast cancer showing heterogeneous HER2 expression. Digital image analysis has emerged as an objective and reproducible scoring method and its role in the assessment of HER2 status with ITH remains to be demonstrated.

## 1. Introduction

Breast cancer is the most commonly diagnosed cancer and the leading cause of death in women worldwide. Globally, the estimated new breast cancer cases reached 2.26 million and accounted for 0.68 million cancer-related deaths in 2020 [1]. In 2023, newly diagnosed female breast cancers and the number of female breast cancer deaths were estimated to be 297,790 and 43,170 in the United States, respectively [2].

The ERBB2 (erb-b2 receptor tyrosine kinase 2) gene is a proto-oncogene encoding a tyrosine kinase receptor, human epidermal growth factor receptor-2 (HER2). HER2 is an orphan receptor with a constitutively activated conformation. It does not possess specific ligand or ligand binding activity. The HER2 protein can form homodimers when HER2 level is high or heterodimers with HER1, HER3 and HER4, but favors HER3. Dimerization of HER2 leads to phosphorylation of its tyrosine kinase domain and activates downstream oncogenic signaling pathways through PI3K/AKT and RAS/RAF/MEK/MAPK/MYC/c-jun to increase cell cycle progression, cell differentiation, survival, angiogenesis, tumorigenesis, migration and invasion [3,4,5]. HER2 overexpression or amplification is identified in approximately 15–20% of breast cancers [6,7] and is associated with higher histologic grade and stage, increased metastatic potential, decreased overall survival (OS), resistance to endocrine therapy and poor response to selected chemotherapy [8,9]. However, the introduction of anti-HER2 targeted therapies has profoundly changed the clinical course of HER2-positive cancer in both early and advanced disease, especially for metastatic breast cancer.

## 2. HER2 Assessment in Breast Cancer

Currently, the main application of HER2 assessment is in prediction of anti-HER2 treatment response in neoadjuvant and adjuvant settings. As a prognostic and predictive biomarker, HER2 status is routinely assessed by immunohistochemistry (IHC) and/or in situ hybridization (ISH) in breast cancer. In order to improve the accuracy of HER2 testing in invasive breast cancer, the American Society of Clinical Oncology and the College of American Pathologists (ASCO/CAP) first drafted a guideline for HER2 testing in breast cancer in 2007 [10]. The guideline recommended that HER2 status should be tested in all newly diagnosed invasive breast carcinoma. Originally, a HER2 positive result was defined as greater than 30% invasive tumor cells with uniform and intense circumferential staining by IHC or *HER2* to centromeric enumeration probe for chromosome 17 (CEP17) ratio > 2.2 or average *HER2* gene copy number greater than six per nucleus by ISH. In order to reduce the small number of potential false-negative results, ASCO/CAP revised the IHC criteria to more than 10% of invasive tumor cells in 2013 [11]. The ISH criteria for HER2 positive were also revised as *HER2*/CEP ratio ≥ 2.0 or *HER2*/CEP ratio < 2.0 with ≥6.0 *HER2* copy number signals per nucleus. Testing was recommended for primary, recurrent and metastatic tumors. The ASCO/CAP further refined the guideline in 2018 [12]. Currently based on the 2018 guideline, breast cancers are classified as either HER2-positive (IHC3+ or 2+ with gene amplification by ISH) or HER2-negative (IHC 0+ or 1+ or 2+ without ISH amplification).

Several clinical trials, including the National Surgical Adjuvant Breast and Bowel Project (NSABP) B-31, the North Central Canter Treatment Group (NCCTG) N9831 and DESTINY-Breast04, demonstrated that a subset of breast cancer patients with HER2 low levels of expression also benefited from anti-HER2 targeted therapy, especially trastuzumab-deruxtecan (T-DXd) [13]. A new concept of “HER2 low” (IHC 1+ or 2+ without ISH amplification) breast cancer has been proposed [14,15,16]. Additional studies in this field may require a shift from the binary HER2 scoring system to a new system including HER2-positive, low and zero (IHC 0). Furthermore, breast cancer with HER2 ultra-low (IHC score 0 with incomplete and faint staining in ≤10% of tumor cells) has been highlighted [17]. The clinical trial investigating the potential benefit of T-DXd in HER2 ultra-low, hormone receptor positive breast cancer in metastatic setting is currently ongoing in DESTINY-Breast06. Currently, the assessment of HER2 low levels of expression in breast cancer has not been formally defined by ASCO/CAP.

## 3. HER2 Intratumoral Heterogeneity

Intratumoral heterogeneity (ITH) is defined as the coexistence of subpopulations of tumor cells that differ genetically, phenotypically or behaviorally within a primary tumor or between a primary tumor and its metastases. ITH poses a remarkable challenge for characterization of biomarkers and treatment selection.

HER2 ITH is a well-known phenomenon in breast cancer. The prevalence of HER2 heterogeneity has been reported in up to 40% of breast cancers [18,19,20,21,22,23,24,25,26,27]. It is rare in HER2 3+ cases, but significantly more common in HER2 equivocal cases [20,21,25,28,29,30]. Several studies also revealed remarkable HER2 ITH in HER2 low status [31].

HER2 ITH is defined by the co-existence of at least two distinct clones of cells with varying HER2 statuses within the same tumor. This means the different areas within the same tumor may have different levels of protein expression or gene amplification. HER2 ITH may present in three distinct patterns based on the geographic (spatial) distribution of heterogeneity: (1) clustered (regional) type, defined as two distinct areas with different *HER2* gene amplified tumor cell populations; (2) mosaic (intermixed) type, defined as diffuse intermingling of cells with different *HER2* gene amplification status; (3) scattered type, defined as isolated *HER2* amplified tumor cells in a predominantly non-amplified tumor [19,32]. The clustered type is reported to be much less common compared to the mosaic type, 0.01% vs. 3% in unselected cohort and 4% vs. 15% of all IHC score 3+ and score 2+ cases subjected to ISH [33]. A genomic study with gene copy number profiling and massively parallel sequencing was investigated in 12 cases with HER2 ITH. It identified potential driver genetic alterations restricted to the HER2-negative cells, suggesting that HER2-negative components are likely driven by genetic alterations not present in the HER2-positive components, including *BRF2* and *DSN1* amplification and *HER2* somatic mutations [34]. The mosaic and scattered types are more frequent and typically encountered in the HER2 2+ and ISH equivocal cases (HER2-double equivocal by 2013 guideline) [33]. Evaluation of HER2 status in these cases could be quite challenging. HER2 gene protein assay (GPA), a new HER2 testing modality with combined HER2 IHC and bright field ISH, enables assessment of HER2 protein expression and gene amplification simultaneously on a single slide and is particularly helpful in cases with HER2 ITH [27,35].

Our recent studies using GPA have suggested HER2 ITH can be classified into two categories based on *HER2* gene amplification and HER2 protein expression in same tumor cells: genetic and non-genetic ITH [26,27,35]. Tumor with genetic ITH shows co-existence of classic HER2-positive tumor cells with both *HER2* gene amplification and HER2 protein overexpression, and HER2-negative tumor cells without *HER2* gene amplification or HER2 protein overexpression. However, tumor with non-genetic ITH harbors tumor cells with *HER2* gene amplification but no HER2 protein overexpression, and classic HER2-positive tumor cells (Figure 1). Tumor cells with such discordant *HER2* gene amplification and HER2 protein expression can only be identified by using GPA, which shows IHC and ISH signals in the same tumor cells on one single slide. The different HER2 ITH patterns are summarized in Table 1.

The difference of HER2 expression levels between primary and residual tumors after neoadjuvant therapies has been reported. The potential mechanism could be that the HER2-positive cells are more sensitive to anti-HER2 targeted therapies which lead to selected proliferation of clones with HER2-negative or low expression [36]. The discordance of HER2 status has also been reported in 0–34% of breast cancers between the primary and metastatic sites [37]. Possible explanations for this wide range of variation could be due to differences in fixation and ischemic time, different patient populations included in study cohorts, different methodologies, and different HER2 antibody clones in different studies. It could also be due to inter-observer interpretation variabilities, especially in tumors with HER2 ITH, as discussed later. The potential mechanisms of loss of HER2 expression in metastases include genetic drift, clonal selection during tumor progression, ITH, aforementioned anti-HER2 therapeutic selection or possible discrepant biomarker testing results [37]. Discordance of HER2 expression between primary and either residual tumor or metastatic settings was associated with poor prognosis or a lack of pathologic complete response (pCR) [37,38].

## 4. ASCO/CAP Guidelines Regarding HER2 ITH

In 2009, ASCO/CAP published a supplement to the 2007 guideline to define HER2 heterogeneity as the presence of 5% to 50% of total tumor cells with *HER2* gene amplification [18]. The 2013 guideline further addressed this issue and recommended reporting HER2 ITH [11]. It defined HER2 heterogeneity as the presence of a second population of cells of which 10% or more were tumor cells with a different *HER2* copy number and/or *HER2*/CEP17 ratio. A separate counting of at least 20 non-overlapping cells within this population must be performed and reported. The 2018 guideline acknowledged that unusual patterns of HER2 expression can be encountered, including strong and complete staining in less than 10% of tumor cells. These heterogeneous cancers would likely benefit from retesting at recurrence and/or metastases [12].

To obtain reliable HER2 IHC results and accurately assess HER2 ITH, pre-analytic, analytic and post-analytic variables need to be standardized. ASCO/CAP HER2 guidelines recommend cold ischemic time should be less than 1 h and the duration of tissue fixation in 10% neutral buffered formalin should be 6–72 h [11]. HER2 IHC can also be negatively affected by acid decalcifying solutions [39,40].

## 5. Clinicopathologic Features of Breast Cancer with HER2 ITH

HER2 ITH exists in HER2-low breast cancers more frequently than HER2-zero or HER2 positive cancers, featuring a pattern of diffuse intermingling of HER2-positive and HER2-negative tumor cells (mosaic pattern) [31,41,42]. The majority of these cases are hormone receptor positive [43]. In contrast to HER2-positive cases, HER2 ITH cases are associated with lower histologic grade, smaller tumor size with a biologic features resembling HER2-negative cases [30]. Among the HER2-negative carcinomas, cases with ITH were associated with larger size, higher grade and greater incidence of lymph node metastasis [29].

## 6. Impact of HER2 Heterogeneity in Anti-HER2 Treatment

Despite significant improvement of survival due to anti-HER2 therapies, around 4–23% of patients with localized disease still experience relapse after anti-HER2 based regimen and become metastatic [44]. Several potential resistance mechanisms to anti-HER2 therapies have been proposed: (1) difficulties in antibody and HER-2 binding: low levels of HER2 expression on the cell surface or modification of the HER2 extracellular binding domain which leads to the impossibility of binding to the HER2 targeted agents; (2) activation of compensatory pathways: for example, increased ER expression, formation of HER2-EGFR heterodimers or alterations of CDK4/6 can stimulate cell growth and survival; (3) alterations in downstream signaling pathways: overexpression of NRG1, constitutive activation of PI3K; (4) escape from antibody-dependent cellular cytotoxic: presence of drug efflux pumps in the cell membrane which reduce the intracellular cytotoxic action of antibody-drug conjugates (ADC), increased ER, constitutive activation of PI3K, alterations of the intracellular binding domain leading to resistance of tyrosine kinase inhibitors [45,46].

Multiple studies indicated that HER2 ITH could be another potential or at least contributive factor for the resistance to anti-HER2 therapy. First, HER2 ITH may contribute to the inaccurate assessment of HER2 status and lead to inappropriate treatment regimens. Indeed, HER2 ITH is a critical contributor of equivocal HER2 testing results and may result in a discordance between IHC and ISH [19,30,32]. HER2 testing performed on a small biopsy of tumor with HER2 ITH may not provide a representative characterization of the tumor as a whole. In cases with HER2 ITH, assessment of the entire slide or multiple slides is recommended. Additionally, areas with different HER2 IHC results should be analyzed by ISH separately. Pathology reports should include the percentage of cells with HER2 IHC 3+ expression and the HER2 status of other populations [47]. This will advise oncologists that tumor areas with different HER2 status may variably respond to anti-HER2 therapy. Second, tumors with HER2 ITH may not bind or uptake anti-HER2 reagents as efficiently as tumors with homogenous HER2 positivity. In the prospective ZEPHIR trial, HER2-positive metastatic breast cancer patients underwent a pretreatment HER2-positron emission tomography (PET)/computed tomography (CT) with (89) Zr-trastuzumab. Intra-patient heterogeneity in HER2 expression was observed in 46% of patients, with nearly one-third of patients having little or no Zr-trastuzumab HER2-PET/CT across their metastatic sites [48]. Furthermore, recent publications have demonstrated that the presence of HER2 ITH was a poor prognostic factor for patients who were treated with HER2-targeted therapies. HER2 ITH was associated with significantly shorter disease-free survival (DFS) and decreased OS, and was less responsive to anti-HER2 targeted therapy [32,49,50,51,52,53]. Our group conducted a retrospective study of anti-HER2 neoadjuvant chemotherapy in a cohort of 64 HER2 positive patients. Our results demonstrated that ITH was present in 30% of patients, and significantly more cases with HER2 ITH were found in the incomplete response group compared to the pCR group (56% vs. 13%, *p* < 0.001), indicating that HER2 ITH is an independent predictor for incomplete response to anti-HER2 neoadjuvant chemotherapy [26]. In a prospective clinical trial, 157 patients with HER2-positive breast cancer were treated with trastuzumab emtansine (T-DM1) in combination with pertuzumab in the absence of conventional chemotherapy. HER2 ITH, defined as an area with *ERBB2* amplification in >5% but <50% or a HER2-negative area in tumor cells by FISH, was detected in 10% of cases. None of the patients with HER2 ITH reached pCR compared to 55% in the non-heterogeneous subgroup. The association between HER2 ITH and pCR remained significant even after being adjusted by ER status [43]. In the T-DM1 plus pertuzumab arm of the KRISTINE trial, 80% of cases showing HER2 ITH among the patients experienced locoregional progression before surgery compared to 15% in patients without locoregional disease progression [54]. In the metastatic setting, the performance of the T-DM1 arms was poor in patients with HER2 ITH (<80% of IHC 2+/3+) compared to patients with homogenous HER2 expression (≥80%) in the MARIANNE trial [52]. A study with *HER2* FISH in single cells demonstrated that the fraction of HER2 non-amplified cells was the driver of resistance [43].

A novel HER2-targeted antibody drug conjugate (ADC), trastuzumab deruxtecan (T-DXd), with a drug to antibody ratio of 7–8, was designed to effectively deliver payload to HER2-expressing cells. The payload is linked to anti-HER2 antibody by a cleavable peptide-based linker. After the linker is cleaved by lysosomal cathepsins in HER2-positive cancer cells, the released drug is cell membrane permeable and can affect cells in close proximity by the bystander effect regardless of their HER2 expression status [55]. In the DESTINY-Breast04 trial, patients with HER2-low metastatic breast cancer treated with T-DXd showed significantly longer progression-free and OS than the physician’s choice of chemotherapy [14]. The T-DXd antitumor activity in heterogeneous or HER2-low expressing tumors may be related to bystander effect or high drug-to-antibody ratio [55,56]. The roles of HER2-targeting therapies, especially with newer developed agents, and the combination with non-targeting agents in breast cancer with HER2 ITH need to be further investigated.

Given the relatively poor response to anti-HER2 targeted therapy in breast cancer patients with HER2 ITH, the treatment regimens can be deescalated or escalated according to the HER2 ITH status. A proposed treatment algorithm based on HER2 ITH status is illustrated in Figure 2. Breast cancer patients with HER2 homogeneous status usually have a higher chance of reaching pCR and show a strong association with longer event-free survival and distant recurrence-free survival. The treatment regimens in this group of patients can be further tuned by other factors. Neoadjuvant anti-HER2 targeted therapy combined with chemotherapy can be applied in ER-negative cases, while anti-HER2 targeted therapy alone may be sufficient for HER2-enriched and PIK3CA wild-type cases. On the other hand, breast cancer patients with HER2 ITH are less likely to reach pCR and subsequently are more likely to develop distant metastases. Escalated treatments with anti-HER2 targeted therapy combined with chemotherapy should be applied from the beginning of the treatment to improve clinical outcomes in this group of patients.

## 7. HER2 Intratumoral Heterogeneity in Other Tumor Types

HER2 overexpression and amplification have been identified in many other tumor types, including colon, stomach, endometrium, ovary, prostate, lung, bladder, etc. [57]. Given the potential therapeutic effect of trastuzumab, HER2 status is routinely assessed in gastric, gastroesophageal junction adenocarcinoma, endometrial serous carcinoma and recently in endometrial carcinosarcoma. Testing is performed by IHC or ISH mirroring the process used in breast carcinoma. Although HER2 ITH is a rare event in breast cancer, it has been documented in a significant percentage of gastric, gastroesophageal junction, bladder, colorectal and lung cancers [29,58,59].

HER2 overexpression was identified in 7–34% of gastric carcinomas/gastroesophageal junction carcinomas [58,60]. HER2-positive patients had shorter survival rates than HER2-negative patients regardless of stage, and HER2 overexpression was significantly associated with OS [57]. HER2 ITH in gastric cancers was reported in 14–79% and 23–54% by IHC and FISH analyses, respectively [60,61,62], which is higher than that in breast cancer. One study reported that HER2 heterogeneous overexpression was significantly associated with longer DFS than homogenous expression, and the high average gene copy number was associated with poor outcome [61]. Other studies found shorter DFS and OS, and poor response to anti-HER2 targeted therapy in HER2 heterogeneous positive groups compared with HER2 homogeneous positive groups. Multivariate analysis stratified by age and chemotherapy revealed that HER2 ITH remained significant [60,62,63]. The DESTINY-Gastric01 trial suggested a clinical benefit of T-DXd in HER2-low gastric cancer patients [64]. So far there is no standard guideline regarding the evaluation of HER2 heterogeneity in gastric cancers. The clinical significance of HER2 ITH in gastric cancer remains to be further evaluated.

HER2 overexpression and gene amplification in endometrial serous carcinoma (ESC) has been reported in 14–80% and 21–47% of cases, respectively [59,65,66,67]. A randomized phase II clinical trial and multiple studies demonstrated antitumor activity of anti-HER2 agents in ESC, endometrial and ovarian carcinosarcoma [68]. It is recommended to routinely assess HER2 status in ESC or mixed carcinoma with a serous component. HER2 ITH was observed in 31–53% of endometrial serous carcinoma [59,69,70]. Trastuzumab as a single agent in GOG study or combined with pertuzumab showed no response in HER2-positive ESC [71,72]. Trastuzumab combined with chemotherapy and trastuzumab duocarmazine showed clinical benefit in HER2-positive ESC [73,74]. The evaluation of efficacy of T-DXd in endometrial carcinosarcoma is currently ongoing. While most of the anti-HER2 agents have been successful in HER2-positve breast cancer, they were not in non-breast HER2-positive solid tumors. One explanation for this difference is the relatively high HER2 heterogeneity within these tumors [75].

## 8. The Possible Role for Artificial Intelligent (AI) in the Assessment of HER2 ITH

Studies revealed significant inter-observer variabilities in assessment of HER2 status in breast cancer with HER2 ITH and HER2 low levels of expression status [31]. In order to accurately assess HER2 status, alternative methods, such as molecular testing or digital pathology, have been proposed [76]. Since the wide implementation of whole slide imaging (WSI), digital image analysis (DIA) has emerged as a quick, cost-effective, objective and reproducible scoring method to assess HER2 status by IHC and anti-HER2 treatment response [77,78,79,80,81,82,83]. Some AI systems were developed to discriminate *HER2* FISH positive from negative cases [84]. The ASCO/CAP HER2 guideline has acknowledged DIA as a diagnostic modality for HER2 status assessment and CAP has created guidelines to facilitate adoption of HER2 DIA into routine pathology workflows [77]. When these guidelines are implemented, there will still be challenging cases particularly in the low to intermediate HER2 expression groups. High inter- and intra-observer variability has been reported in multiple studies, particularly in HER2 IHC 2+ cases [55,85,86]. Additional studies have suggested AI could reduce HER2 IHC equivocal cases and increase interpretation accuracy and interobserver agreement [78,79,87,88], but still performed relatively poorly in cases showing heterogeneous staining patterns. The newly developed AI-assisted microscope combines the advantages of pathologists and AI, where the pathologists can provide real-time feedback to AI and adjust the AI results. This has significantly improved the accuracy of AI results [84]. Taken together, the application and reliability of AI in assessing cases with HER2 low levels of expression, particularly with HER2 ITH is yet to be determined.

## 9. Conclusions

HER2 intratumoral heterogeneity is a well-documented phenomenon in multiple tumor types. In this review, we provided an overview of HER2 ITH in breast cancer. HER2 ITH exists in approximately 10–40% of breast cancers and makes it difficult to accurately evaluate HER2 status, accordingly, to determine the appropriate treatment strategy. HER2 ITH has been proven to be associated with poor prognosis, shorter disease-free survival, decreased overall survival and less response to traditional anti-HER2 targeted therapy. Given the promising efficacy of new anti-HER2 therapy of antibody-drug conjugates (T-DXd, etc.) in HER2-low breast cancers, it is necessary to explore HER2 ITH in these tumors and its effect on the efficacy of antibody-drug conjugates. Further understanding of HER2 heterogeneity within a tumor can be important to predict response to anti-HER2 targeted therapies, guide treatment decision, improve clinical outcomes and shed light on development of novel anti-HER2 agents to overcome resistance.

## Figures and Tables

**Figure 1 cancers-15-02664-f001:**
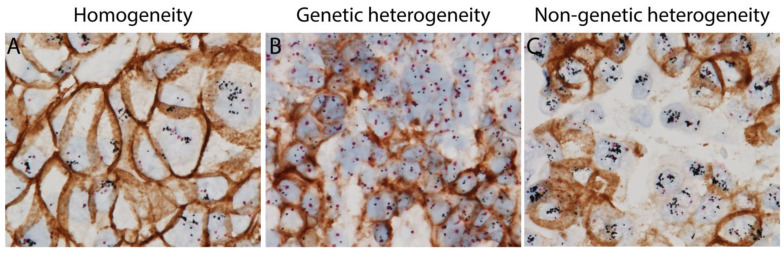
HER2 homogenous and heterogeneous staining patterns (intratumoral heterogeneity (ITH)) using HER2 gene protein assay (GPA) with simultaneous HER2 in situ hybridization and immunohistochemistry on breast cancer tissue sections. (**A**) Homogeneous staining with classic HER2 positive tumor cells with both amplified *HER2* gene and overexpressed HER2 protein. (**B**) HER2 genetic intratumoral heterogeneity with classic HER2 positive tumor cells and HER2 negative tumor cells. (**C**) HER2 non-genetic intratumoral heterogeneity with a mixture of classic HER2 positive tumor cells and non-classic HER2 positive tumor cells harboring amplified *HER2* gene, but no overexpression of HER2 protein.

**Figure 2 cancers-15-02664-f002:**
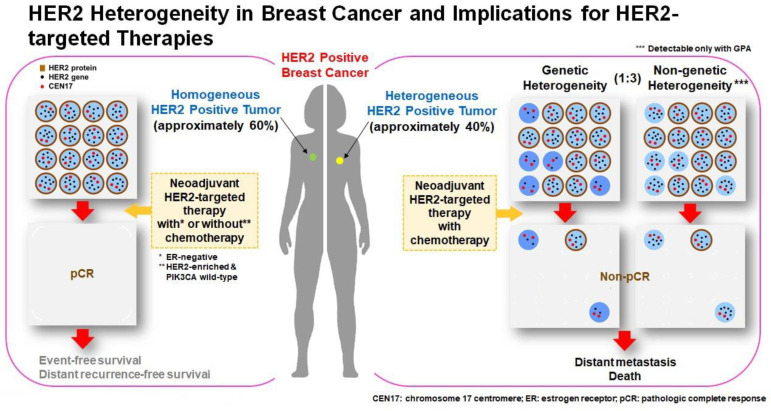
HER2 heterogeneity in breast cancer and implications for HER2-targeted therapies. Approximately 60% of HER2-positive breast cancer is homogeneous regarding HER2 status in invasive tumor cells (**left**) and 40% shows HER2 intratumoral heterogeneity either with genetic heterogeneity or non-genetic heterogeneity (**right**). Breast cancer patients with HER2 homogeneous status have a higher chance of reaching pCR, longer event-free survival and distant recurrence-free survival. The treatment regimens can be further tuned by other factors. Neoadjuvant anti-HER2 targeted therapy combined with chemotherapy can be applied in ER-negative cases, while anti-HER2 targeted therapy alone may be sufficient for HER2-enriched and PIK3CA wild-type cases. Breast cancer patients with HER2 ITH are less likely to reach pCR and develop distant metastases. Escalated treatments with anti-HER2 targeted therapy combined with chemotherapy should be applied from the beginning of the treatment to improve clinical outcomes in this group of patients.

**Table 1 cancers-15-02664-t001:** Summary of HER2 intratumoral heterogeneity patterns.

HER2 ITH Patterns		Definition
Genetic	Clustered Type	Two distinct areas with different *HER2* gene amplification
	Mosaic Type	Diffuse intermingling of cells with different *HER2* amplification status
	Scattered Type	Isolated *HER2* amplified tumor cells in a predominantly non-amplified tumor
Non-genetic		Tumor cells with *HER2* gene amplification without HER2 protein expression intermixed with tumor cells with concordant HER2 amplification and protein expression

## Data Availability

Not applicable.

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
