# Peer review of "HER2 Intratumoral Heterogeneity in Breast Cancer, an Evolving Concept"

_cancers, 2023, doi:10.3390/cancers15102664_

Round 1

Reviewer 1 Report

Well written comprehensive article that bones in on an important concept of HER2 heterogeneity and the impact on treatment outcomes.

Author Response

Thanks for your professional review work and comments.

Reviewer 2 Report

This review addresses an important, yet poorly defined and little understood aspect of HER2 diagnosis – intratumoral heterogeneity (ITH). Overall, the manuscript is divided into well organized sections. The figures are clearly labeled and comprehensible. However, the grammar and language usage could be improved.

Specific points:

1)      In Introduction, “ligand-bling” should read “ligand binding.”

2)      In Section 3. HER2 intratumoral heterogeneity, the last paragraph is not clear.  How could anti-HER2 treatments “CAUSE decreased HER2 amplification and subsequently reduced protein expression in the same tumor cell population?” If the authors mean that the treatments cause selection of tumor cells with these traits, how does this differ from the following potential mechanism? The next statement that “the discordance of HER2 status has also been reported in 0-34% breast cancers between the primary and metastatic sites” suggests that there is disagreement among studies, but no references are supplied, and the discrepancy is not discussed.

3)      In Section 6. Impact of HER2 heterogeneity in anti-HER2 treatment, the phrase “on the cell wall” in the first paragraph should read “in the cell membrane.” In the 4th paragraph, the sentence, “On the other hand, breast cancer patients with HER2 ITH is less likely to reach pCR and subsequently develop distant metastasis” should be modified to “On the other hand, breast cancer patients with HER2 ITH are less likely to reach pCR and subsequently more likely to develop distant metastases.”

4)      In Section 7. HER2 intratumoral heterogeneity in other tumor types, the sentence in the 2nd paragraph that reads, “Lee et al. reported the HER2 heterogeneous overexpression was significantly associated with LONGER DFS than the homogenous…” should be checked for accuracy as it appears to conflict with clinical data cited subsequently. Also, it is not clear what the word “retained” means in the sentence, “Multivariate analysis stratified by age and chemo-therapy revealed that HER2 ITH RETAINED to be significant.”

Some editing is necessary for clarity.

Author Response

Specific points:

1) In Introduction, “ligand-bling” should read “ligand binding.”

We changed it to “ligand binding” according to the reviewer’s suggestion.

2) In Section 3. HER2 intratumoral heterogeneity, the last paragraph is not clear.  How could anti-HER2 treatments “CAUSE decreased HER2 amplification and subsequently reduced protein expression in the same tumor cell population?” If the authors mean that the treatments cause selection of tumor cells with these traits, how does this differ from the following potential mechanism? The next statement that “the discordance of HER2 status has also been reported in 0-34% breast cancers between the primary and metastatic sites” suggests that there is disagreement among studies, but no references are supplied, and the discrepancy is not discussed.

We appreciate the reviewer’s comments. The exact mechanism of difference of HER2 expression levels between primary and residual tumors is unclear. We tried to give some examples. We agree with the reviewer that there is no evidence to support the statement that the same group of cells had decreased HER2 amplification and subsequently protein expression. Therefore, we deleted that sentence.

As for the 0-34%, it was from our previous publication which was listed as reference 37. In order to decrease the No. of references, we did not include all the original articles which are listed below.

Ref 37: Discordance rates for HER2 of 0-34% between primary breast cancer and its paired metastatic tumor have been reported [4,5,8,10].

4 Ataseven B, Gologan D, Gunesch A, et al:HER2/neu, topoisomerase 2a, estrogen and progesterone receptors: discordance between primary breast cancer and metastatic axillary lymph node in expression and amplification characteristics. Breast Care (Basel) 2012;7:465–470.

5  Rossi S, Basso M, Strippoli A, et al: Hormone receptor status and HER2 expression in primary breast cancer compared with synchronous axillary metastases or recurrent metastatic disease. Clin Breast Cancer 2015;15: 307–312

8  Nishimura R, Osako T, Okumura Y, Tashima R, Toyozumi Y, Arima N: Changes in the ER, PgR, HER2, p53 and ki-67 biological markers between primary and recurrent breast cancer: discordance rates and prognosis. World J Surg Oncol 2011;9:131.

10 Liedtke C, Broglio K, Moulder S, et al: Prognostic impact of discordance between triplereceptor measurements in primary and recurrent breast cancer. Ann Oncol 2009;20:1953–1958.

We added the discussion according to the reviewer’s suggestion. “The discordance of HER2 status has also been reported in 0-34% breast cancers between the primary and metastatic sites [37]. The possible explanations of these wide ranges of variation could be due to differences in fixation and ischemic time, different patient populations included in study cohorts, different methodologies, and different HER2 antibody clones in different studies. It could also be due to inter-observer interpretation variabilities, especially in tumors with HER2 ITH as discussed later.”

3)      In Section 6. Impact of HER2 heterogeneity in anti-HER2 treatment, the phrase “on the cell wall” in the first paragraph should read “in the cell membrane.” In the 4th paragraph, the sentence, “On the other hand, breast cancer patients with HER2 ITH is less likely to reach pCR and subsequently develop distant metastasis” should be modified to “On the other hand, breast cancer patients with HER2 ITH are less likely to reach pCR and subsequently more likely to develop distant metastases.”

       We kindly appreciated the reviewer’s suggestion. Both sentences were modified correspondingly.

4) In Section 7. HER2 intratumoral heterogeneity in other tumor types, the sentence in the 2ndparagraph that reads, “Lee et al. reported the HER2 heterogeneous overexpression was significantly associated with LONGER DFS than the homogenous…” should be checked for accuracy as it appears to conflict with clinical data cited subsequently. Also, it is not clear what the word “retained” means in the sentence, “Multivariate analysis stratified by age and chemo-therapy revealed that HER2 ITH RETAINED to be significant.”

In order to draw the conclusion that the clinical significance of HER2 ITH in gastric cancer remains unclear and needs further evaluation, we tried to give some examples which provided contrary results. We reviewed the original article by Lee et al., the data is accurate. We rewrote this section as below.

“One study reported the HER2 heterogeneous overexpression was significantly associated with longer DFS than the homogenous, and the high average gene copy number was associated with poor outcome [61]. While other studies found shorter DFS and OS, poor response to anti-HER2 targeted therapy in HER2 heterogeneous positive groups compared with HER2 homogeneous positive groups.

We also changed “retained” to “remained” to reduce potential confusion.

Reviewer 3 Report

The authors have reviewed HER2 intratumoral heterogeneity in breast cancer in this paper.   Understanding HER2 heterogeneity within a tumor can be important to predict response to anti-HER2 targeted therapies, guide treatment decisions, improve clinical outcomes and shed light on developing novel anti-HER2 agents to overcome resistance.  This review is systematic and rich in content with complete references, and the study conforms to the scope of the journal;   the reviewing methods and conclusion are accurate.   Therefore, I think that this manuscript is worth publication.

Author Response

We greatly appreciate the reviewer's professional work and comments. 

Reviewer 4 Report

- PDF page 2: to provide the full name of the abbreviation "CEP".

- figure 2: it seems it is, as indicated, from a publication from JAMA Oncology 2020; 6:1355-62. Is it used with the permission from JAMA Oncology?

Author Response

  1. We thanks for the reviewer's suggestion. We added "centromeric enumeration probe for chromosome 17" (CEP17).
  2. As for figure 2, Dr. Nitta created that figure. We cited the JAMA oncology article as a reference since the I-SPY2 trial confirmed that there is a strong association of pCR with EFS and DRFS based on a 3-year follow-up. A longer follow-up period will sure provide additional information.